



# Brief communication: Sampling c-axes distributions from the eigenvalues of ice fabric orientation tensors

Martin Rongen[1]

[1]RWTH Aachen University, Institute for Particle Physics III B, 52074 Aachen, Germany

**Correspondence:** Martin Rongen (rongen@physik.rwth-aachen.de)

**Abstract.** For simulation purposes involving different realizations of ice fabrics, it can be necessary to generate arbitrarily large samples of c-axes based on the second-order orientation tensor, a commonly used descriptive statistics provided in publications of ice core measurements. This paper describes a sampling technique based on the combination of a vertical girdle and a single maximum Watson distributions.

## 1 Introduction

On Earth, water ice usually occurs as a polycrystal of the Ih crystal structure. Each monocrystal is weakly birefringent with the optical axis coinciding with the c-axis of the crystal. The viscosity of an individual crystal depends strongly on the direction of the applied strain. As hexagonal crystal, ice will most readily deform as shear is applied orthogonal to the c-axis leading to slip of the individual basal planes.

As described in Alley (1988), convergent flow results in c-axes which follow a preferential distribution, called a girdle fabric, where most c-axes lie on a plane perpendicular to the flow direction. In contrast, basal shear, divergent flow and parallel flow cause c-axes to rotate toward the vertical axis, yielding a unimodal distribution. For more context see for example Faria et al. (2014).

For simulation purposes, such as the simulation of light being diffused while propagating through the crystal structure of
polar ice as described in Chirkin and Rongen (2019), arbitrarily large samples of c-axes from different, realistic fabrics may be required.

As measured c-axis distributions always only offer limited statistics and are restricted to the encountered fabric states, it can be necessary to statistically sample such generic c-axis distribution.

To be able to compare to glaciological literature, the input to the sampling shall be the second-order orientation tensor, which
is commonly used in this field to summarize the characteristics of measured ice fabrics.





## 2 The orientation tensor

One can describe the N c-axes, measured in an ice sample, by N unit vectors $\boldsymbol{n}_i$, with components $n_{ix}, n_{iy}, n_{iz}$. Note that $\boldsymbol{n}_i$ is equivalent to $-\boldsymbol{n}_i$ as the vector can be chosen to point along either direction of the axis. Scheidegger (1965) suggested to represent this ensemble of vectors via the matrix:

$$
5 \quad a = \begin{bmatrix} \sum n_{ix}^2 & \sum n_{ix} \cdot n_{iy} & \sum n_{ix} \cdot n_{iz} \\ \sum n_{iy} \cdot n_{ix} & \sum n_{iy}^2 & \sum n_{iy} \cdot n_{iz} \\ \sum n_{iz} \cdot n_{ix} & \sum n_{iz} \cdot n_{iy} & \sum n_{iz}^2 \end{bmatrix} \tag{1}
$$

The normalized form $A = a/N$ is called the second order orientation tensor. It was introduced in glaciology through Gödert and Hutter (1998). $A$ has three eigenvectors and three corresponding eigenvalues $S_1$, $S_2$ & $S_3$, with $S_1 + S_2 + S_3 = 1$.

The axes of the coordinate system in which the c-axes are evaluated can be chosen such that the x-axis points along the mean c-axis direction $\sum \boldsymbol{n}_i/N$, that the z-axis points along the pole to the best-fit girdle to the distribution (see Woodcock (1977)) and that the y-axis is orthogonal to the other two. In this case the coordinate axes are the eigenvectors, $S_j = \sum_i n_{ij}^2$ and the eigenvalues follow a strict ordering such that $S_1 > S_2 > S_3$.

A perfectly uniform, girdle or unimodal fabric features the following relations between the eigenvalues:

– uniform:      $S_1 \approx S_2 \approx S_3 \approx 1/3$

– unimodal:      $S_1 \approx 1; \quad S_2 \approx S_3 \approx 0$

– girdle:      $S_1 \approx S_2 \approx 0.5; \quad S_3 \approx 0$

Woodcock (1977) realized that all possible fabric states can be visualized in a 2D plot, as only two of the three eigenvalues are independent. He suggested the representation, as shown in Figure 1, where the abscissa is given as $ln(S_2/S_3)$ and the ordinate is given as $ln(S_1/S_2)$.

In this representation uniform c-axis distributions are found at the origin of the plot. The distance from the origin $C = ln(S_1/S_3)$ is called the strength parameter. Girdle fabrics are found to the lower-right, while unimodal fabrics reside to the upper-left. The type of fabric can also be quantified by the so called Woodcock shape parameter $\kappa = ln(S_1/S_2)/ln(S_2/S_3)$. Large $\kappa$ values denote a unimodal fabric. $\kappa$ values smaller than 1 denote a gridle fabric.

Glaciological papers usually denote the measured ice fabric by directly publishing the eigenvalues or by presenting their data as a scatter plot in the coordinate system defined by Woodcock. See for example Donald E., Voigt, (2017).

## 3 Sampling approach

Neither the orientation tensor nor its eigenvalues retain the full information on the ensemble of underlying c-axes. Thus, an assumption on the functional form of the fabric has to be made when trying to sample a distribution. Here we focus on describing random, girdle and unimodal distributions, as well as combinations of these, as those are the types most commonly encountered in ice fabric measurements.





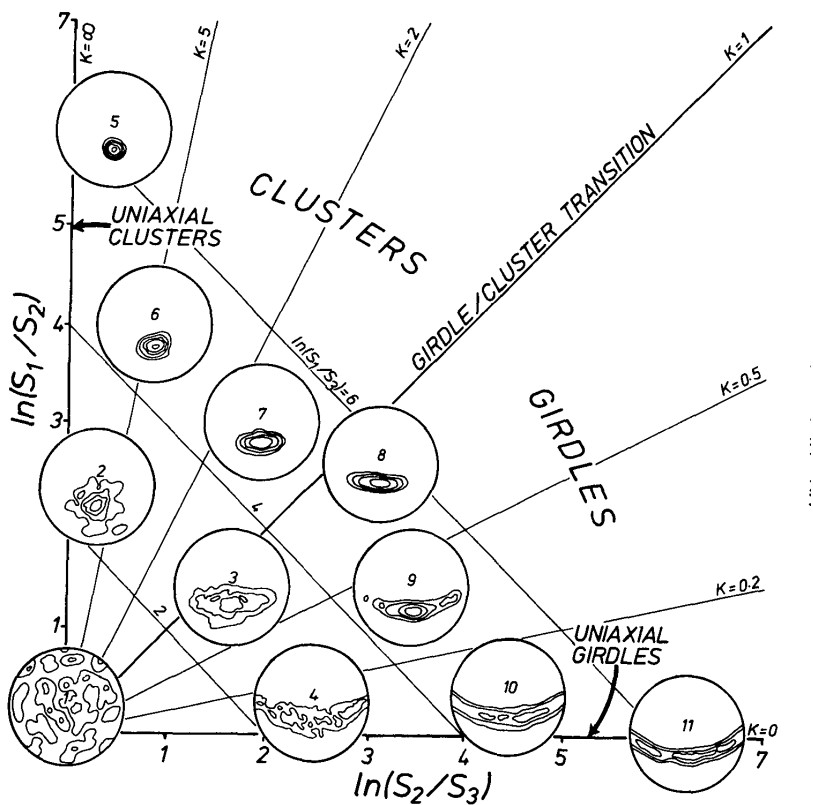

**Figure 1.** Two-axis logarithmic plot of ratios of eigenvalues $S_1$, $S_2$ & $S_3$, with examples of fabric shapes in different parts of the graph. [Woodcock (1977)]

The book "Statistical analysis of spherical data" by Fisher et al. (1987) gives a good overview of commonly used probability density functions (PDFs) for directional data. Of the presented PDFs the Watson (1965) distribution seems most applicable for our case as:

1. It can represent both unimodal as well as rotational symmetric girdle data.
2. There exists an (approximate) parameter estimation based on eigenvalues alone.

In its standardized form the PDF has only one free parameter $k$ and is given as:

$$f(\theta, \phi) = C_w \exp(k \cdot \cos^2 \theta) \sin \theta \tag{2}$$

with $\quad C_w = 1 / \left( 4\pi \int_0^1 exp(k \cdot u^2) du \right)$. In the following the Python package available at https://github.com/duncandc/watson_distribution is used to sample from the Watson distribution. Alternatively the sampling approach described in Fisher et al. (1987) may be used.





Figure 2 shows example distributions for different values of $k$. At $k = 0$ the direction distribution is perfectly uniform. For negative $k$ the distribution is bimodal in vector space which is equivalent to a unimodal distribution in axis space and has the highest probability at the poles. For positive $k$ values the distribution is girdle with the directions equally distributed around the equator.

$$k < 0 \qquad\qquad k = 0 \qquad\qquad k > 0$$

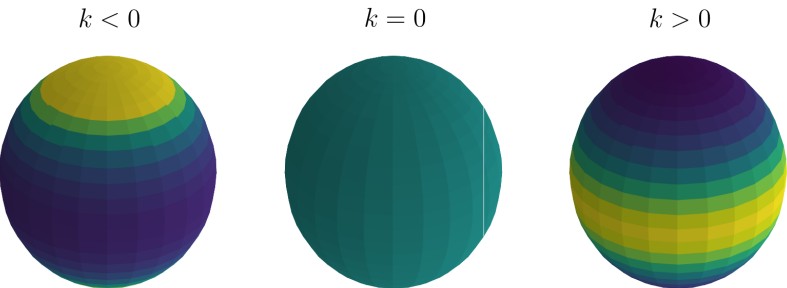

**Figure 2.** Example Watson distributions
[D. Campbell, https://github.com/duncandc/watson_distribution]

5  Best and Fisher (1986) showed that, for a purely girdle or unimodal distribution, the $k$-parameter can be estimated from the eigenvalues as follows.

$$k = \begin{cases} 3.75 \cdot (3 \cdot S_1 - 1), & \frac{1}{3} \leq S_1 \leq 0.34 \\ -5.95 + 14.9 S_1 + \frac{1.48}{1 - S_1} - \frac{11.05}{S_1^2}, & 0.34 < S_1 \leq 0.64 \\ -7.96 + 21.5 \cdot S_1 + \frac{1}{1 - S_1} - 13.25 \cdot S_1^2, & S_1 > 0.64 \end{cases} \tag{3}$$

for a unimodal fabric and for a girdle fabric

$$k = \begin{cases} \frac{1}{2 \cdot S_3}, & 0 \leq S_3 \leq 0.06 \\ 0.961 - 7.08 \cdot S_3 + \frac{0.466}{S_3}, & 0.06 < S_3 \leq 0.32 \\ 3.75 \cdot (1 - 3 \cdot S_3), & 0.32 < S_3 \leq \frac{1}{3} \end{cases} \tag{4}$$

10  For ice fabrics, the plane of girdle c-axes shall intersect the poles, where also the c-axes of a unimodal distribution are found. As such the directions sampled from girdle Watson distributions are rotated by $90°$. Due to the underlying rotational symmetry the eigenvalues of the resulting Watson distributions follow a strict relation:

$$S_1 = S_2 \quad \& \quad S_3 = 1 - 2 \cdot S_1 \quad \text{, for a girdle Watson}$$

$$S_2 = S_3 \quad \& \quad S_1 = 1 - 2 \cdot S_2 \quad \text{, for a unimodal Watson} \tag{5}$$

15  Obviously no single Watson distribution can describe an arbitrary set of eigenvalues with $S_1 \neq S_2 \neq S_3$. This is achieved by combining directions sampled from a girdle and a unimodal Watson distribution.

Given a sample of c-axes from a girdle Watson distribution with eigenvalues $S_{ig}$ and a sample of c-axes from a unimodal Watson distribution with eigenvalues $S_{iu}$, as well as a relative fractional contribution of the girdle sample $f_g$ to the total sample,



the eigenvalues of the combined sample $S_i$ are given by: $S_i = f_g \cdot S_{ig} + (1 - f_g) \cdot S_{iu}$ The combination of $f_g$, $S_{1g}$ & $S_{2u}$ which yields the desired eigenvalues $S_1$, $S_2$ & $S_3$ is found by solving the equation system $S_i$, which has been simplified using the relations in equation 5:

$$S_1 = f_g \cdot S_{1g} + (1 - f_g) \cdot (1 - 2 \cdot S_{2u}),$$

$$S_2 = f_g \cdot S_{1g} + (1 - f_g) \cdot S_{2u},$$

$$S_3 = f_g \cdot (1 - 2 \cdot S_{1g}) + (1 - f_g) \cdot S_{2u} \tag{6}$$

The third equation is not independent as $S_1 + S_2 + S_3 = 1$. Thus further information is needed to be able to constrain the variables. To fulfill the assumption that $S_u$ is unimodal and $S_g$ is girdle one can further constrain $1/3 < S_{1g} < 0.5$ and $0 < S_{2u} < 1/3$. For cases, where the system is still underconstrained, one can for example further demand that $S_{2g} = S_{2u}$

(equivalent to $S_{1u} + S_{3g} = 1$), so that both distributions have an equal spread around the gridle plane. The solution is then given as:

$$f_g = 0.5 \cdot (\epsilon - 4S_1 - 2S_2 + 3)$$
$$S_{1g} = \frac{2S_1}{-\epsilon + 4S_1 + 2S_2 + 1}$$
$$S_{2u} = \frac{\epsilon - 2S_2 - 1}{2 \cdot (\epsilon - 4S_1 - 2S_2 - 1)} \tag{7}$$

with: $\epsilon = \sqrt{16 \cdot S_1^2 + 16S_1 \cdot (S_2 - 1) + (2S_2 + 1)^2}$.

From these one can derive the Watson parameters $k$ using the approximations as given in equations 3 and 4.

## 4 Examples and discussion

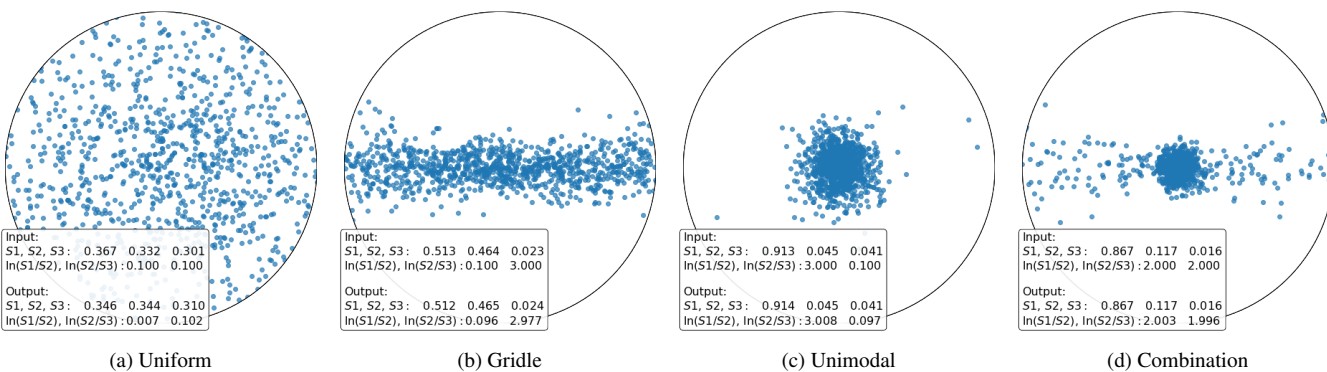

(a) Uniform (b) Gridle (c) Unimodal (d) Combination

**Figure 3.** Example c-axis distributions generated using the described method.

To verify and visualize the success of the presented sampling approach, c-axes distributions according to a number of

combinations of $ln(S_1/S_2)$ & $ln(S_2/S_3)$ have been generated as seen in Figure 3. The sampled c-axes distributions yield





eigenvalues which are accurate to within the approximation of the parameter estimation for the Watson distributions and well sufficient for most applications.

Note that by design the c-axes distributions for intermediate fabric states do not contain a single elliptical distribution but a rotationally symmetric girdle and a circular unimodal. This seems suitable for our application to ice fabrics. In very deep

5    glacial ice where the fabric slowly evolves from girdle to unimodal, experimental distributions such as published by Weikusat et al. (2017) indeed show the described superposition and not an elliptical distribution usually sketched for these eigenvalues. Still care has to be taken when applying the presented method, so to be assured that the axis distribution to be reproduced does indeed fulfill the requirement to be able to be described by the superposition of a purely gridle and a purely unimodal Watson distribution. This is very much the case for example for the SPICEcore data (Donald E., Voigt, (2017)), but may not

10    necessarily be the case as for example seen in the multi-modal distributions found by Wilson and Peternell (2011).

*Competing interests.* The authors declare that they have no competing interests (both financial or non-financial).

*Acknowledgements.* This work was supported in part by BMBF, Verbundforschung and was carried out in the context of the IceCube Neutrino Observatory.



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
