# Peer review of "Brief communication: Sampling c-axes distributions from the eigenvalues of ice fabric orientation tensors"

_The Cryosphere, 2019_

## Referee Comment (RC1) · Anonymous Referee #1 · 4 Dec 2019

This brief communication presents a sampling method to build discrete c-axis distribution for given eigenvalues of the second-order orientation tensor using a superposition of girdle and single maximum fabric given as a Watson distribution.

Even if the proposed method is certainly interesting, the glaciology context is clearly missing. A number of previous works have already been done on that subject and are not mentioned in this paper. For example:
- in Gagliardini et al. (2009), a comprehensive list of the PDFs that have been proposed in the literature to describe polycrystalline fabric is given and they are all

compared in their capability of representing observed fabrics.

- the form proposed in this paper was already proposed as a good possible representation of ice fabric by Lliboutry in 1993.

- Gillet-Chaulet et al. (2005), with the same objective as in the current paper, have presented a method to construct a discrete fabric for given eigenvalues of the second orientation tensor assuming a parameterised PDF derived from an analytical solution and capable to describe directly orthotropic fabrics (without the need of superposition of two PDFs restricted to transversally isotropic fabrics as in the proposed approach).

In other words, there are clearly missing references to give an appropriate context of what have already been done on that subject in glaciology (and certainly other than the three listed here).

Over the three figures, Fig. 1 is from Woodcock (1977) and Fig. 2 is from Duncan Campbell (from his github). Did the author get the authorization to replicate these figures in his paper?

Regarding the result of the method, I don't really understand why the uniform fabric presented in Fig. 3a has not $S_1 = S_2 = S_3 = 1/3$ exactly. This should be possible? It is surprising that it is for the simplest fabric (uniform) that there is the largest differences between the input and output eigenvalues. As discussed at the end of the paper, it seems that the assumption that a natural fabric can be described by the superposition of a purely girdle and a purely unimodal Watson distribution is a bit strong and doesn't work especially for not textured fabric (i.e. fabric close to a uniform distribution). I would have like to see an inverse approach showing how a real fabric can be described using the superposition of two transversally isotropic PDFs, as done in Gagliardini et al. (2009).

Minor remarks:

- page 1, line 8: polycrystalline ice will most

- page 2, line 11: I don't understand the "strict" ordering (eigenvalues can be all equal for a uniform distribution). It should write $S_1 \geq S_2 \geq S3$.

- page 2, line 24: the citation should be Voigt (2017). Donald is the first name, not the family name. Same at other places and in the references section.

Gagliardini O., F. Gillet-Chaulet and M. Montagnat, 2009. A Review of Anisotropic Polar Ice Models: from Crystal to Ice-Sheet Flow Models. In "Physics of Ice Core Records II", Supplement Issue of Low Temperature Science, Vol. 68, December 2009.

Gillet-Chaulet F., O. Gagliardini , J. Meyssonnier, M. Montagnat and O. Castelnau, 2005. A user-friendly anisotropic flow law for ice-sheet modelling. J. of Glaciol., 51(172), p. 3-14.

L. Lliboutry. Anisotropic, transversely isotropic non linear viscosity of rock ice and rheological pa- rameters inferred by homogenization. Int. J. Plast., 9:619–632, 1993.

---

## Referee Comment (RC2) · Wataru Shigeyama (Referee) · 27 Jan 2020

This brief communication describes a method to generate arbitrary c-axis orientations from eigenvalues of ice fabric orientation tensors. I understood that the novelty of this study is to combine girdle and unimodal Watson distributions to generate the arbitrary c-axis orientations. Although there is a limitation in a specific case, the method can produce c-axis orientation distributions similar to those with input eigenvalues. However, the author should rewrite or reconsider the manuscript for the publication of The Cryosphere. The main reasons are the following.

1. There have been several methods for describing arbitrary c-axis orientation distri-

bution of ice (e.g., Seddik et al., 2008; Gagliardini et al., 2009). The author should write appropriate research backgrounds and point out current problems. This could better convey the importance of the study, namely, combining two Watson distributions to sample c-axis orientations.

2. Following the references the author cited in the manuscript (Fisher et al., 1987 and Best and Fisher, 1986), the distribution is bipolar (unimodal) if the parameter $\kappa$ (kappa) is positive, while the distribution is girdle if the kappa parameter is negative. The kappa parameter in the manuscript is k in equation (2). The relationship between k and the distributions (unimodal and girdle), described in page 4 line 1 and in Figure 2, is opposite to what is written in the references (Fisher et al., 1987 and Best and Fisher, 1986). Why are they different? I have to suspect that the estimation of k is appropriate because the estimation is based on a study of Best and Fisher (1986).

Minor comments and corrections

1. Do the words "sampling c-axes distribution" (e.g., in the title, page5-line20) express what the author would like to? One would usually say "sampling c-axes (or c-axis orientations)" and then obtaining those distributions which he or she assumes the population of c-axis orientations. If this comment is not appropriate, please ignore it.

2. The author could write about the examples of c-axis orientation sampling (Section 4) in the Abstract. Some spaces remain there.

3. "N" should be written with italic style as it is variable (page 2-line 2).

4. The author could consider replacing "&" with "and" as use of "&" is not official in some cases (e.g., page 2-line 7; page 3-figure caption 1; page5-line1, -line2, -line20).

5. "ln" should be written with block style as it is not variable (page 2-line17, -line18, line21; page 5-line20).

6. The citation of Donald E., Voigt (2017) may not be a very appropriate example because Donald E., Voigt (2017) does not show scatter plots of the fabric data in the

Woodcock's coordinate system. The author could consider adding a more appropriate reference.

7. The author should explain the variables in equation (2) briefly (page3-line7), maybe together with a coordinate system, and show the coordinate system (axes) in Figure 2 and 3.

8. "exp" in equation (2) should be written with block style as it is not variable (page3-line7).

9. The third term in the second equation in equation (3) may be mistyping (page5-line3).

10. The author should consider showing the derivation of equation (7) briefly (page5-line14). It would be more reader-friendly.

11. "BMBF" should be spelled out, or its meaning should be translated to English (page6-line12).

References Seddik, H., Greve, R., Placidi, L., Hamann, I., & Gagliardini, O. (2008). Application of a continuum-mechanical model for the flow of anisotropic polar ice to the EDML core, Antarctica. Journal of Glaciology, 54(187), 631-642. Gagliardini, O., Gillet-Chaulet, F., Montagnat, M., & Hondoh, T. (2009). A Review of Anisotropic Polar Iee Models: from Crystal to Ice-Sheet Flow Models in "Physics of ice core records II (ed. Hondoh, T.)", 149-166.

I hope the above comments are useful. Wataru Shigeyama

---

## Author Comment (AC1) · 27 Feb 2020

Dear Wataru Shigeyama,

Thank you for your timely and detailed review. Please find the responses to the issues raised in-line with your review comments below:

This brief communication describes a method to generate arbitrary c-axis orientations

[Figure]

from eigenvalues of ice fabric orientation tensors. I understood that the novelty of this study is to combine girdle and unimodal Watson distributions to generate the arbitrary c-axis orientations. Although there is a limitation in a specific case, the method can produce c-axis orientation distributions similar to those with input eigenvalues. However, the author should rewrite or reconsider the manuscript for the publication of The Cryosphere. The main reasons are the following.

1. There have been several methods for describing arbitrary c-axis orientation distribution of ice (e.g., Seddik et al., 2008; Gagliardini et al., 2009). The author should write appropriate research backgrounds and point out current problems. This could better convey the importance of the study, namely, combining two Watson distributions to sample c-axis orientations.

> While this brief communication article was intended as a minimalistic method description, I appreciate that some context about previous work in the field is needed. Based also on additional comments from the other referee, the following section shall be added to the introduction:

> "Probability density distribution functions (PDFs) previously used to sample c-axes have for example been summarized in Gagliardini et al. (2009). These include most prominently the Fisher distribution and its restricted form as proposed for use in glaciology by Lliboutry (1993). Both distributions are only applicable when describing a single maximum and can not be used for axially symmetric girdle fabrics. A single Watson distribution, to describe either a girdle or a unimodal fabric, has already been successfully used for example by Kennedy et al. (2013). Gillet-Chaulet et al. (2005) presented a conjugate gradient method which, starting from an arbitrary c-axis

distribution, in-time converges to a distribution that can describe any orthotropic fabric. While generally applicable, the conjugate gradient method is computationally inefficient and as such undesirable for the intended purpose. As a result, we present a method based on sampling the combination of a vertical girdle and a single maximum Watson distribution, which is computationally efficient and can reproduce arbitrary eigenvalues of the second order orientation tensor."

In addition the introduction of the Watson distribution in section 3, page 3 shall be extended to:

"Of the presented PDFs the Watson (1965) distribution, as also for example used by Kennedy et al. (2013), seems most applicable for our case as ..."

2. Following the references the author cited in the manuscript (Fisher et al., 1987 and Best and Fisher, 1986), the distribution is bipolar (unimodal) if the parameter $\kappa$(kappa) is positive, while the distribution is girdle if the kappa parameter is negative. The kappa parameter in the manuscript is k in equation (2). The relationship between k and the distributions (unimodal and girdle), described in page 4 line 1 and in Figure 2, is opposite to what is written in the references (Fisher et al., 1987 and Best and Fisher,1986). Why are they different? I have to suspect that the estimation of k is appropriate because the estimation is based on a study of Best and Fisher (1986).

Thank you for pointing out this inconsistency! The source code I have been using defined the Watson distribution with an ... $\exp(-k \cdot ...$, in contrast to the original references and equation (1). The manuscript has been updated for consistency (so

that negative values result in a girdle distribution).

In answering this question, I also realized that the uses of $\kappa$ for the Woodcock shape parameter and k for the Watson distribution are inconsistent with most of the previous literature. To avoid potential confusion the use will be changed so that $\kappa$ is the parameter of the Watson distribution and the capital K is the Woodcock shape parameter.

Minor comments and corrections:

(a) Do the words "sampling c-axes distribution" (e.g., in the title, page 5 - line 20) express what the author would like to? One would usually say "sampling c-axes (or c-axis orientations)" and then obtaining those distributions which he or she assumes the population of c-axis orientations. If this comment is not appropriate, please ignore it.

> Changing the title to "Sampling c-axes from the eigenvalues of ice fabric orientation tensors" makes the title more concise and will be adopted for the revised manuscript.

(b) The author could write about the examples of c-axis orientation sampling (Section 4) in the Abstract. Some spaces remain there.

> This is a nice suggestion and will be added to the manuscript: "... This paper describes a sampling technique based on the combination of a vertical girdle and a single maximum Watson distribution. A number of application examples is provided."

(c) "N" should be written with italic style as it is variable (page 2 - line 2).

> This will changed to the referee's recommendation.

(d) The author could consider replacing "&" with "and" as use of "&" is not official in some cases (e.g., page 2 - line 7; page 3 - figure caption 1; page 5 - line 1, - line 2, - line 20).

> This will changed to the referee's recommendation.

(e) "ln" should be written with block style as it is not variable (page 2-line17, -line18,line21; page 5-line20).

> This will changed to the referee's recommendation.

(f) The citation of Donald E., Voigt (2017) may not be a very appropriate example because Donald E., Voigt (2017) does not show scatter plots of the fabric data in the Woodcock's coordinate system. The author could consider adding a more appropriate reference.

> The SP14 reference was chosen here, as the presented method was developed to describe fabrics akin to the once found in SP14. A reference to the WAIS Divide Ice Core (DOI: 10.3189/2014JoG14J100) will be added, which contains both a Woodcock diagram and fabrics which can be described by the described method.

(g) The author should explain the variables in equation (2) briefly (page 3 - line 7), maybe together with a coordinate system, and show the coordinate system (axes) in Figure 2 and 3.

> The unmentioned variables $\theta, \phi$ are the polar (colatitude) and azimuth angle of the standard spherical coordinate system. This will be mentioned in the revised manuscript: "In its standardized form the PDF, evaluated on a spherical coordinate system with the polar angle $\theta$ and the azimuth angle $\phi$, has only one free parameter .... with the

normalization constant $C_w$ ..."

(h) "exp" in equation (2) should be written with block style as it is not variable (page 3 - line 7).

> This will be changed to the referee's recommendation.

(i) The third term in the second equation in equation (3) may be mistyping (page 5 - line 3).

> Please elaborate. I double-checked the equation with the Best & Fisher publication and my source code and it seems fine.

(j) The author should consider showing the derivation of equation (7) briefly (page 5 - line 14). It would be more reader-friendly.

> Going from equation (6) to equation (7) simply requires solving an equation system with two equations and two unknowns. To highlight this, the implication that the assumption $S_{2g} = S_{2u}$ also means $S_{2g} = S_{2u} = S_{1g}$ will be added to the manuscript.

(k) "BMBF" should be spelled out, or its meaning should be translated to English (page 6 - line 12).

> This will changed to the referee's recommendation. It's the "Bundesministerium für Bildung und Forschung" or "Federal Ministry of Education and Research".

References Seddik, H., Greve, R., Placidi, L., Hamann, I., and Gagliardini, O. (2008). Application of a continuum-mechanical model for the flow of anisotropic polar ice to the EDML core, Antarctica. Journal of Glaciology, 54(187), 631-642.

Gagliardini, O.,Gillet-Chaulet, F., Montagnat, M., and Hondoh, T. (2009). A Review of Anisotropic Polar lee Models: from Crystal to Ice-Sheet Flow Models in "Physics of ice core records II (ed. Hondoh, T.)", 149-166.

---

## Author Comment (AC2) · 27 Feb 2020

Dear Referee,

Thank you for your timely and detailed review. Please find the responses to the issues raised in-line with your review comments below:

This brief communication presents a sampling method to build discrete c-axis distribu-

[Figure]

tion for given eigenvalues of the second-order orientation tensor using a superposition of girdle and single maximum fabric given as a Watson distribution.

Even if the proposed method is certainly interesting, the glaciology context is clearly missing. A number of previous works have already been done on that subject and are not mentioned in this paper. For example:

- in Gagliardini et al. (2009), a comprehensive list of the PDFs that have been proposed in the literature to describe polycrystalline fabric is given and they are all compared in their capability of representing observed fabrics.

- the form proposed in this paper was already proposed as a good possible representation of ice fabric by Lliboutry in 1993.

- Gillet-Chaulet et al. (2005), with the same objective as in the current paper, have presented a method to construct a discrete fabric for given eigenvalues of the second orientation tensor assuming a parameterised PDF derived from an analytical solution and capable to describe directly orthotropic fabrics (without the need of superposition of two PDFs restricted to transversally isotropic fabrics as in the proposed approach).

In other words, there are clearly missing references to give an appropriate context of what have already been done on that subject in glaciology (and certainly other than the three listed here).

> While this brief communication article was intended as a minimalistic methods description, I appreciate that some context about previous work in the field is needed. As the referee may have recognized, my primary occupation is not in glaciology, so I am thankful for

the provided references. Working through them I see the following context:

Gagliardini et al. (2009) summarizes a number of commonly used PDFs, including the original work by Lliboutry. It shall be mentioned to provide general context. The Watson distribution is not explicitly mentioned in this publication.

The Lliboutry (1993) paper proposes the use of a Fisherian distribution and of a distribution given by $f_\nu(\theta) = \nu \cos^{\nu-1} \theta$. It does not mention a Watson distribution or combinations of Watson distributions and as such does not include the form proposed in this paper.

The Gillet-Chaulet et al. (2005) paper is very interesting and indeed pursues the same objective. Yet in using a conjugate gradient method to converge from an initial (arbitrary) c-axis distribution to a distribution resulting in the desired eigenvalues is computationally inefficient. In the application for which the presented method was developed (photon propagation through a birefringent polycrystal), millions of c-axes need to be generated on the fly, rendering the method described by Gillet-Chaulet et al. inapplicable.

Branching out from the provided references, the Kennedy et al. (2013, doi: 10.3189/2013JoG12J159) publication is a nice example of a previous use of a single Watson distribution to generate crystal distributions. The innovation in the presented manuscript shall

be understood as the proper analytic combination of two Watson distributions, so to yield arbitrary eigenvalues of the second order orientation tensor.

Based also on additional comments from the other referee, the following section shall be added to the introduction:

> "Probability density distribution functions (PDFs) previously used to sample c-axes have for example been summarized in Gagliardini et al. (2009). These include most prominently the Fisher distribution and its restricted form as proposed for use in glaciology by Lliboutry (1993). Both distributions are only applicable when describing a single maximum and can not be used for axially symmetric girdle fabrics. A single Watson distribution, to describe either a girdle or a unimodal fabric, has already been successfully used for example by Kennedy et al. (2013). Gillet-Chaulet et al. (2005) presented a conjugate gradient method which, starting from an arbitary c-axis distribution, in-time converges to a distribution which can describe any orthotropic fabric. While generally applicable, the employed method is computationally inefficient and as such undesirable for the intended purpose. As a result we present a method based on the combination of a vertical girdle and a single maximum Watson distributions, which is computationally efficient and can reproduce arbitrary eigenvalues of the second order orientation tensor."

In addition the introduction of the Watson distribution in section 3, page 3 shall be extended to:

> Of the presented PDFs the Watson (1965) distribution, as
> also for example used by Kennedy et al. (2013), seems most
> applicable for our case as ...

Over the three figures, Fig. 1 is from Woodcock (1977) and Fig. 2 is from Duncan
Campbell (from his github). Did the author get the authorization to replicate these
figures in his paper?

> Thank you for pointing out this issue. I shall discuss the two figures
> with the editor and replace or drop them.

Regarding the result of the method, I don't really understand why the uniform fabric
presented in Fig. 3a has not $S_1 = S_2 = S_3 = 1/3$ exactly. This should be possible? It is
surprising that it is for the simplest fabric (uniform) that there is the largest differences
between the input and output eigenvalues.

> The fabric input which is to be recovered in Figure 3a is setup to
> not be perfectly uniform, but to be $\ln(S1/S2) = \ln(S2/S3) = 0.1$,
> which is very closely recovered. Attached please find a figure which
> indeed reproduces an ideal uniform fabric. As it seems the chosen
> example for Figure 3a has lead to some confusion, the pure uniform
> fabric shall be adopted for the new manuscript revision.

As discussed at the end of the paper, it seems that the assumption that a natural fabric
can be described by the superposition of a purely girdle and a purely unimodal Watson

distribution is a bit strong and doesn't work especially for not textured fabric (i.e. fabric close to a uniform distribution).

> When aiming for a perfectly uniform distribution ($S_1 = S_2 = S_3 = 1/3$) the k-parameter calculation for the the Watson distribution (equations (3) and (4) will yield exactly 0 in both cases. And the sampled distribution will be exactly uniform in the limit of infinite statistics. Also see the answer above and the attached figure.

I would have like to see an inverse approach showing how a real fabric can be described using the superposition of two transversally isotropic PDFs, as done in Gagliardini et al. (2009).

> To be honest I do not see much added value from such a test. The limitations of the methods should be clear at that point and the result of the comparison (as for example done in Figure 2 of Gagliardini et al. (2009)) will primarily depend on the chosen dataset (as the functional form of the PDF is fixed).

Minor remarks:

- page 1, line 8: polycrystalline ice will most

  > This will be changed to the referee's recommendation.

- page 2, line 11: I don't understand the "strict" ordering (eigenvalues can be all equal for a uniform distribution). It should write $S_1 \geq S_2 \geq S_3$.

  > They will never be numerically identical in any natural system.

[Figure]

> But to also include this edge case, I will change this to the referee's recommendation.

- page 2, line 24: the citation should be Voigt (2017). Donald is the first name, not the family name. Same at other places and in the references section.

> Indeed. This will be corrected.

Gagliardini O., F. Gillet-Chaulet and M. Montagnat, 2009. A Review of Anisotropic Polar Ice Models: from Crystal to Ice-Sheet Flow Models. In "Physics of Ice Core Records II", Supplement Issue of Low Temperature Science, Vol. 68, December 2009.
Gillet-Chaulet F., O. Gagliardini, J. Meyssonnier, M. Montagnat and O. Castelnau,2005. A user-friendly anisotropic flow law for ice-sheet modelling. J. of Glaciol., 51(172), p. 3-14.
L. Lliboutry. Anisotropic, transversely isotropic non linear viscosity of rock ice and rheological parameters inferred by homogenization. Int. J. Plast., 9:619–632, 1993.

[Figure]

Input:
$S1, S2, S3$ :   0.333   0.333   0.333
$\ln(S1/S2), \ln(S2/S3)$ : 0.000   0.000

Output:
$S1, S2, S3$ :   0.334   0.333   0.333
$\ln(S1/S2), \ln(S2/S3)$ : 0.003   $-0.000$

**Fig. 1.**